# Spatiotemporal biocontrol and rhizosphere microbiome analysis of Fusarium wilt of banana

Zhiyan Zhu[1,2], Guiyun Wu[3,4], Rufang Deng[5], Xiaoying Hu[5], Haibo Tan[4], Yaping Chen[2], Zhihong Tian [1✉] & Jianxiong Li [6✉]

The soil-borne fungus *Fusarium oxysporum* f. sp. *cubense* tropical race 4 (*Foc* TR4) causes Fusarium wilt of banana (FWB), which devastates banana production worldwide. Biocontrol is considered to be the most efficient approach to reducing FWB. Here we introduce an approach that spatiotemporally applies *Piriformospore indica* and *Streptomyces morookaensis* strains according to their respective strength to increase biocontrol efficacy of FWB. *P. indica* successfully colonizes banana roots, promotes lateral root formation, inhibits *Foc* TR4 growth inside the banana plants and reduces FWB. *S. morookaensis* strain Sm4-1986 secretes different secondary compounds, of which xerucitrinin A (XcA) and 6-pentyl-α-pyrone (6-PP) show the strongest anti-*Foc* TR4 activity. XcA chelates iron, an essential nutrient in pathogen-plant interaction that determines the output of FWB. 6-PP, a volatile organic compound, inhibits *Foc* TR4 germination and promotes banana growth. Biocontrol trials in the field demonstrated that application of *S. morookaensis* lead to improvement of soil properties and increase of rhizosphere-associated microbes that are beneficial to banana growth, which significantly reduces disease incidence of FWB. Our study suggests that optimal utilization of the two biocontrol strains increases efficacy of biocontrol and that regulating iron accessibility in the rhizosphere is a promising strategy to control FWB.

[1] College of Life Sciences, Yangtze University, Jingzhou, China. [2] South China Botanical Garden, Chinese Academy of Sciences, Guangzhou, China. [3] Science and Technology Innovation Center, Guangzhou University of Chinese Medicine, Guangzhou, China. [4] Key Laboratory of Plant Resources Conservation and Sustainable Utilization, South China Botanical Garden, Chinese Academy of Sciences, Guangzhou, China. [5] Public Laboratory of Sciences, South China Botanical Garden, Chinese Academy of Sciences, Guangzhou, China. [6] Guangxi Key Laboratory of Agro-environment and Agric-products safety, College of Agriculture, Guangxi University, Nanning, China. ✉email: zhtian@yangtzeu.edu.cn; jxli920@gxu.edu.cn

Bananas (*Musa* spp.), originated in Southeast Asia and the Western Pacific[1,2], are now widely distributed throughout the humid tropics and sub-tropics where they provide a staple food for about 400 million people in the developing countries in Africa, Asia, and Latin America[3]. Bananas are the most exported fruit in the world, having a production of 119 million tons and export trade value of 13.3 billion dollars in 2020[4].

Fusarium wilt of banana (FWB) caused by the soil-borne fungus *Fusarium oxysporum* f. sp. *cubense* (*Foc*) is one of the most destructive diseases in banana production worldwide. FWB has restricted banana production for more than a century[5]. The epidemic of FWB led to the replacement of *Foc* race 1 susceptible Gros Michel with resistant Cavendish, which currently covers about 40% of the global production and is likely only banana present on supermarket shelves for non-producing countries[3,6]. However, a newly emerged race of *Foc*, tropical race 4 (TR4), is virulent not only on Cavendish but also on almost all other banana cultivars. *Foc* TR4 caused serious losses in banana plantation worldwide, which resulted in the abandonment of thousands of hectares of banana orchards[7]. Currently, there are no effective methods to control FWB caused by *Foc* TR4.

The *Foc* pathogen can linger in soil for up to 30 years even in the absence of plant hosts, which makes it particularly difficult to be eliminated from the infected soil[8,9]. Being a vascular pathogen, *Foc* colonizes banana roots and reaches the vascular bundles[3], leading to the ineffectiveness of chemical control. Repeated use of chemical fungicides has raised great concern for environmental pollution and human health. Breeding resistant cultivars is thought to be the most effective way to control FWB, but all commercial banana varieties tested are susceptible to *Foc* TR4, and they are propagated by cloning due to the nature of sterile triploid[10]. Thus, biological control of FWB has gained great interest[11].

*Piriformospora indica* is a well-known endophytic fungus that colonizes roots of a broad spectrum of plant species and confers diverse beneficial effects on host plants[12–14]. *P. indica* colonization of banana significantly alleviated disease symptoms caused by *Foc* TR4[15], and if the colonization was combined with the addition of *Dictyophorae echinovolvata* culture substrates, the best disease resistance enhancement to *Foc* TR4 infection was achieved[16]. *Streptomyces* are naturally abundant in soils, and it is likely that they will cause less damage to the surrounding eco-system when applied. Many *Streptomyces* species have been used as biocontrol agents to protect plants against various diseases due to their ability to produce a range of secondary metabolites that can either inhibit the growth of phytopathogens or promote plant growth[17–19]. *Streptomyces malaysiensis* 8ZJF-21 was isolated from medicinal plants in Hainan of China. The extracts of *S. malaysiensis* 8ZJF-21 showed strong inhibitory effects on the mycelial growth of *Foc* TR4 and when the strain was used as a biocontrol agent, it significantly reduced disease symptoms of the banana plantlets infected by *Foc* TR4[20]. *Streptomyces* sp. H3-2 was isolated from the rhizosphere of banana plantations. The extracts of the strain H3-2 suppressed the growth and spore germination of *Foc* TR4, and pot experiment showed that *Streptomyces* sp. H3-2 promoted the vegetative growth of banana plantlets and significantly inhibited the Fusarium wilt disease symptom caused by *Foc* TR4[21]. Additionally, the application of *Streptomyces* improved the soil microbial communities and enhanced plant resistance to pathogens[22]. Rhizosphere is an important interface involved in the exchange of resources including nutrients, compounds, etc. between the plants, the soil environment, and the microbes. It has been known that competition for essential nutrients such as iron in the rhizosphere is a crucial factor that determines the survival of microbes[23]. Rhizosphere microorganisms that produce growth-inhibitory siderophores could suppress the pathogen growth and thus protect plants against pathogen infection[24]. The approach that takes advantage of the ability of microbes in restricting pathogens access to the essential nutrients is an effective strategy for biocontrol.

Here, we present a spatiotemporal approach using *Piriformospore indica* in the endophytic compartments and *Streptomyces morookaensis* in the rhizosphere to control FWB. *S. morookaensis* strain 4-1986 (Sm4-1986) improves soil properties, shifts rhizosphere microbial structures, and suppresses *Foc* TR4 growth by secreting active compounds when applied to the field. *P. indica* colonizes and proliferates in the intracellular spaces within the banana roots to promote lateral root formation and restrict the growth and extension of *Foc* TR4 inside the banana roots. In addition, soil properties such as pH and iron content are also important factors affecting the control of FWB.

## Results

**Identification of metabolic compounds suppressing *Foc* TR4.** *S. morookaensis* Sm4-1986 displayed antifungal activity against *Foc* TR4[25]. We reasoned that the anti-*Foc* TR4 activity of Sm4-1986 might be due to the secreted compounds. In PDB medium, GFP-labeled *Foc* TR4 grew well showing strong fluorescence (Fig. 1a), however, the addition of the Sm4-1986 supernatant significantly suppressed the growth of *Foc* TR4 as revealed by the great reduction of GFP intensity (Fig. 1b), suggesting the antifungal activity of Sm4-1986 supernatant against *Foc* TR4.

A set of metabolic compounds was isolated from Sm4-1986 supernatant and tested for the antifungal activity[26]. Xerucitrinin A (XcA) (Fig. 1c) and 6-pentyl-α-pyrone (6-PP) (Fig. 1d) are particularly interesting because they showed the strongest anti-*Foc* TR4 activity. Further analysis showed that 3 mM XcA completely suppressed the germination of GFP-labeled *Foc* TR4 spores and greatly reduced GFP fluorescence when compared with the untreated GFP-labeled *Foc* TR4 as examined by laser confocal microscopy (Supplementary Fig. 1a, b). Since Sm4-1986 was able to produce iron-chelating siderophores[25], we then examined the iron-chelating activity of the isolated compounds. Chrome Azurol Sulphonate (CAS) agar assay showed that there was a clear yellow zone around the Oxford cup, indicating a strong iron-chelating ability of XcA (Fig. 1e).

6-PP is another important compound isolated from Sm4-1986. *Foc* TR4 grew well in PDB medium as indicated by the consistently increasing optical density, however, the optical density of *Foc* TR4 culture decreased when 6-PP (0.96 mM final concentration) were added (Fig. 1f), suggesting the inhibition of 6-PP on the growth of *Foc* TR4. To further investigate the effect of 6-PP on *Foc* TR4 growth, we counted the number of *Foc* TR4 spores under microscope. 6-PP treatment significantly reduced the number of *Foc* TR4 spores and completely inhibited spore germination (Fig. 1g). Consistent with this, confocal observation showed that GFP fluorescence was greatly diminished after 6-PP treatment (Supplementary Fig. 1c). *Foc* TR4 is able to produce fusaric acid that acts as a virulent factor to increase environmental acidity[27]. We observed the pH value of *Foc* TR4 solution under normal condition was greatly decreased after 72 h cultivation, however, the pH value of *Foc* TR4 solution treated by 6-PP did not change even after cultivation for 96 h (Supplementary Fig. 1d).

To explore morphological and structural changes of *Foc* TR4 spores under the treatments of XcA and 6-PP, we used scanning electron microscope (SEM) and transmission electron microscope (TEM) to observe the shape of *Foc*TR4 spores. The surface of *Foc* TR4 spores became wrinkled when treated with XcA (Fig. 2a) and the spores were collapsed when treated with

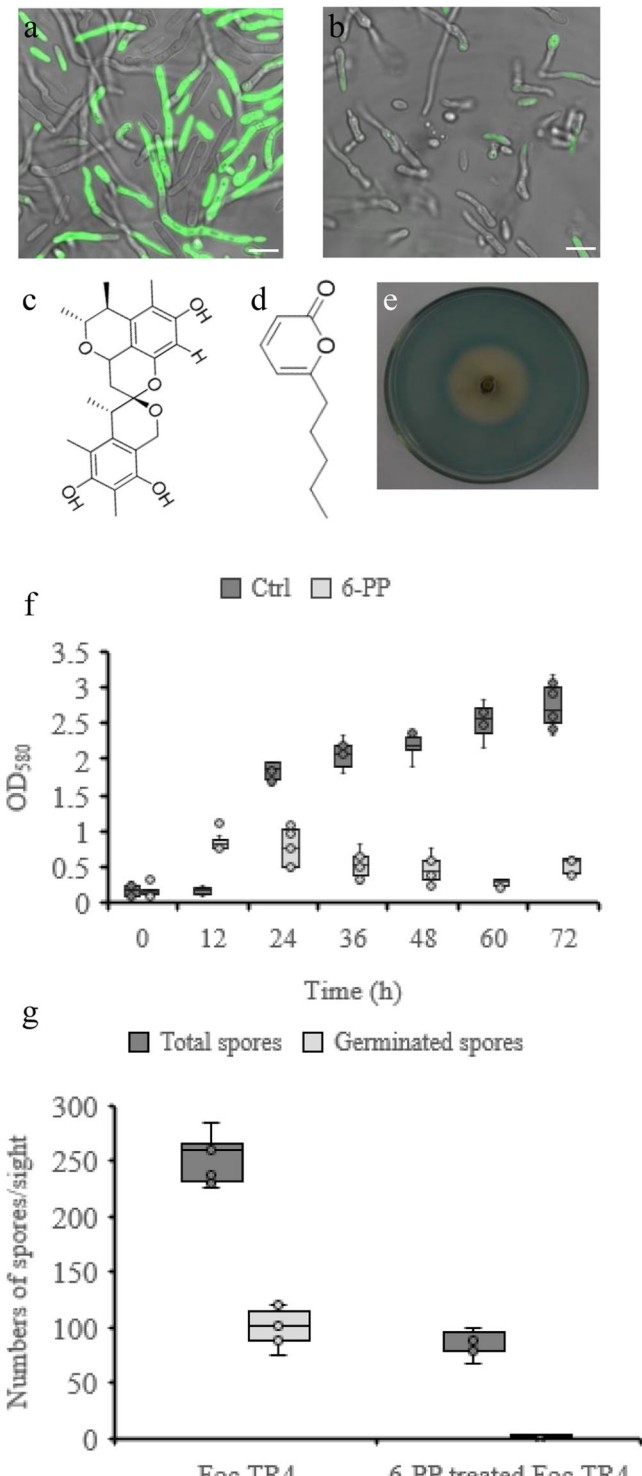

**Fig. 1 Antifungal activity of *S. morookaensis* strain Sm4-1986 against *Foc* TR4. a** GFP-tagged *Foc* TR4 cultured in PDB for 36 h and photographed. **b** The supernatant of *S. morookaensis* strain Sm4-1986 inhibits growth of *Foc* TR4. Strain Sm4-1986 was cultured in PDB for 10 days, and the supernatant of culture was collected and added to PDB (1:1, v/v), in which GFP-tagged *Foc* TR4 was cultured for 36 h and photographed. **c** Structure of xerucitrinin A. **d** Structure of 6-pentyl-α-pyrone. **e** Xerucitrinin A chelates iron assayed by Chrome Azurol Sulphonate (CAS). There is yellow zone around the Oxford cup in the center of the plate, indicating the iron-chelating ability. **f** 6-PP suppresses propagation of *Foc* TR4. *Foc* TR4 and 0.96 mM 6-PP-treated *Foc* TR4 were cultured in PDB, respectively, and the optical densities of the culture solution were monitored at the indicated time-points. Mean of triplicate and standard deviation were shown. **g** 6-PP treatment significantly reduces the number of *Foc* TR4 spores and completely inhibits *Foc* TR4 spore germination. Spores were counted using a Malassez hemocytometer at 24 h of cultivation. Mean of triplicate and standard deviation was shown. Scare bars in (**a**), and (**b**), 5 μm.

XcA and 6-PP and to examine how these two compounds synergistically affect *Foc* TR4 germination at different concentrations. RSM analyses revealed the coexistence of XcA and 6-PP at lower concentrations was able to suppress *Foc* TR4 spore germination, indicating a synergistic effect of these two compounds (Fig. 2h and Supplementary Fig. 2). Given the inhibitory effect of XcA and 6-PP on *Foc* TR4, we investigated their effects on banana growth. Experiments with banana plantlets showed that lower concentrations of 6-PP (< 150 μM) promoted banana plantlet growth although higher concentrations (>200 μM) inhibited banana plantlet growth and caused brown color on the rhizomes (Supplementary Fig. 3). With regard to XcA, banana plantlets grew well when treated for 65 days with 3 mM XcA, the same concentration that inhibited *Foc* TR4 germination, when compared with the control. This indicates that XcA may be non-toxic to banana plants (Supplementary Fig. 4).

**P. indica induces lateral root formation and suppresses *Foc* TR4 growth.** *P. indica* is symbiotically associated with a variety of host plants[28]. To explore the colonization pattern of *P. indica* in banana, we observed the *P. indica*-treated banana roots under microscope. *P. indica* penetrated banana roots primarily through root hairs (Supplementary Fig. 5a) and, later, crossed cortex and endodermis, and moved to stele and aggregated at the lateral root primordium initiation sites (Fig. 3a and Supplementary Fig. 5b). In agreement with these phenomena, *P. indica*-treated banana plantlets showed more lateral roots than untreated ones (Fig. 3b, c).

*Foc* TR4 is able to penetrate cortical parenchyma cells to reach the vascular bundle tissues of roots[29]. Therefore, inhibiting *Foc* TR4 growth and extension in the endophytic compartments of banana roots is an important part of FWB control. To investigate the interaction between *Foc* TR4 and *P. indica*, we co-cultured these two strains and observed their overlaid hyphae. SEM images showed that *P. indica* tightly clasped *Foc* TR4 and resulted in the collapse of *Foc* TR4 hyphae, suggesting an inhibitory effect of *P. indica* on *Foc* TR4 (Fig. 3d–f).

We also examined the effects of *P. indica* on growth of banana and control of Fusarium wilt disease. Inoculation with *P. indica* (1 ×10^6 chlamydospores/ml) increased the growth of banana plantlets (Supplementary Fig. 6a, b and Supplementary Table 1). However, infection with *Foc* TR4 led to the occurrence of typical Fusarium wilt disease symptoms on banana plantlets (Supplementary Fig. 6c). Nonetheless, when banana plantlets were first inoculated with *P. indica* and then infected with *Foc* TR4, less disease symptoms were observed and grew better than *Foc* TR4-inoculated plantlets (Supplementary Fig. 6c, d and Supplementary

6-PP solution (Fig. 2b) compared to the normal *Foc* TR4 spores (Fig. 2c). Furthermore, SEM observation showed that treatment with 6-PP for 24 h resulted in a thinner cell wall and a massive cytoplasm without integral organelles such as mitochondria (Fig. 2d, e) when compared with the normally grown *Foc* TR4 spores (Fig. 2f, g).

Since XcA and 6-PP were isolated from Sm4-1986 and each has antifungal activity against *Foc* TR4, we hypothesized that the coexistence of these two compounds may have synergistic effect on suppressing *Foc* TR4. To test this hypothesis, we used response surface methodology (RSM) to analyze the interaction between

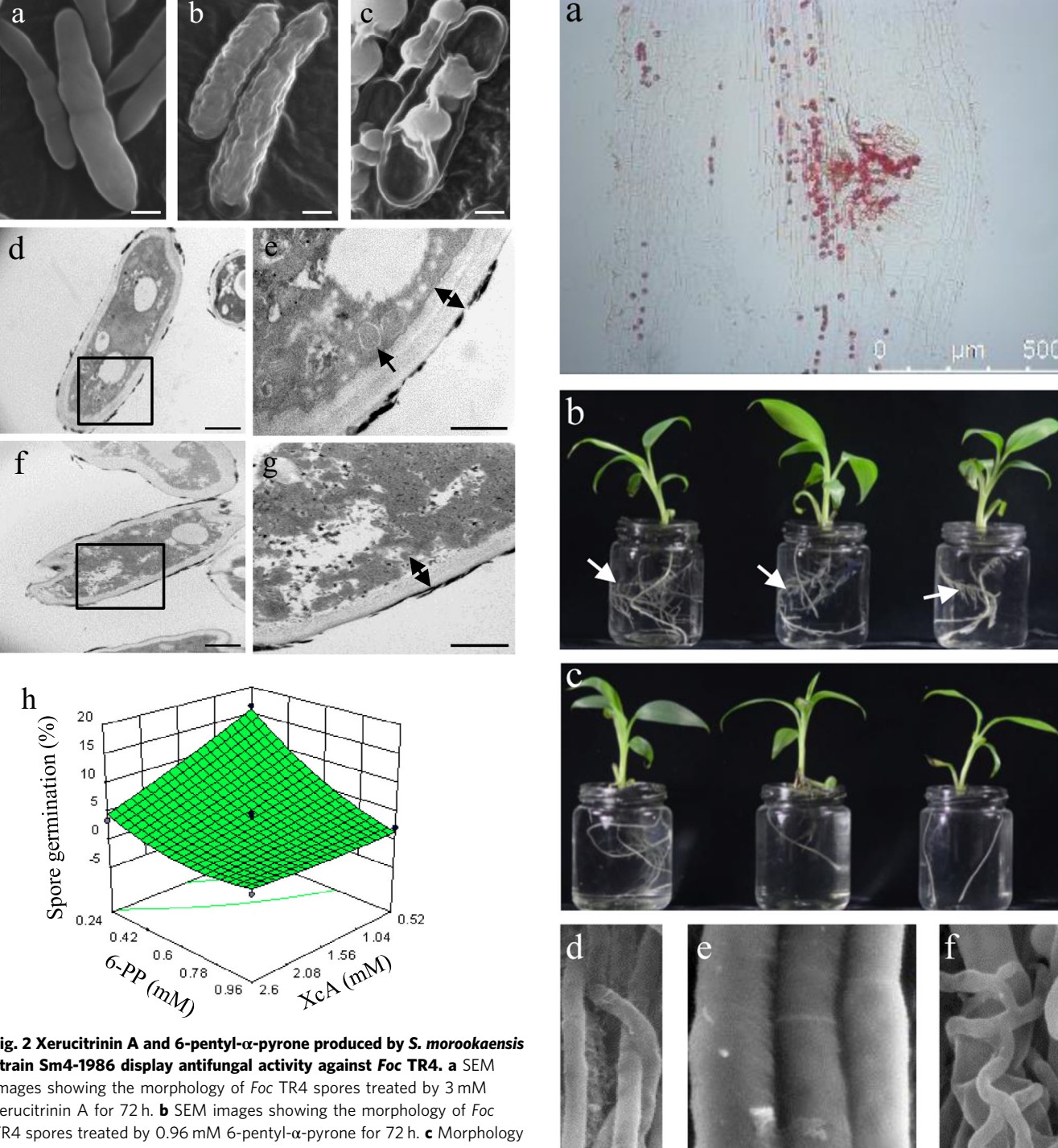

**Fig. 2 Xerucitrinin A and 6-pentyl-α-pyrone produced by *S. morookaensis* strain Sm4-1986 display antifungal activity against *Foc* TR4. a** SEM images showing the morphology of *Foc* TR4 spores treated by 3 mM xerucitrinin A for 72 h. **b** SEM images showing the morphology of *Foc* TR4 spores treated by 0.96 mM 6-pentyl-α-pyrone for 72 h. **c** Morphology of the normal *Foc* TR4 spores analyzed by SEM. **d** Transmission electron microscope (TEM) image of the *Foc* TR4 spores treated by 0.96 mM 6-pentyl-α-pyrone for 24 h. **e** Magnification of the squared area in (**d**) showing a massive cytoplasm without integral organelles. The double arrow indicates the thinned cell wall. **f** TEM image of the normal *Foc* TR4 spores. **g** Magnification of the squared area in (**f**) showing mitochondrion (indicated by arrows). The double arrow indicates the thickness of cell wall. **h** Response surface methodology analysis shows interaction effect of xerucitrinin A and 6-pentyl-α-pyrone on antifungal activity against *Foc* TR4. Scare bars in (**a**–**d**), and (**f**), 1 μm; and in (**e**) and (**g**), 500 nm.

**Fig. 3 *P. indica* stimulates lateral root formation in banana plants and restricts *Foc* TR4 growth in vitro. a** *P. indica* chlamydospores colonize banana roots and aggregate at the lateral root primordium site. **b** Banana plantlets treated by *P. indica* inoculum ($1 \times 10^5$ chlamydospores/ml) and grown in ½ Hoagland medium exhibit more lateral roots. Arrows indicate lateral roots. **c** Untreated banana plantlets grown in ½ Hoagland medium as controls. **d**, **e** Hypha size of *P. indica* (**d**) and *Foc* TR4 (**e**) under scanning electron microscope. **f** *P. indica* hyphae clasp and collapse the hyphae of *Foc* TR4. Scale bars, 2 μm.

Table 1). In agreement with the external symptoms, investigation of the internal symptoms of banana rhizomes showed that *P. indica* treatment greatly reduced the discoloration caused by *Foc* TR4 infection (Supplementary Fig. 6e–h).

**P. indica and S. morookaensis synergistically control FWB in field.** Experiments in greenhouse demonstrated that both *P. indica* and *S. morookaensis* are able to promote banana growth and suppress *Foc* TR4[25]. We further investigated the biocontrol efficacy of these two strains in field.

The field trials were carried out for two consecutive years in a land that had been abandoned due to the severe infection of *Foc* TR4. The field and bananas plantlets were treated with Sm 4-1986 and *P. indica*, respectively, 7 days before transplantation. By the end of the year, banana plants in the field were scored for Fusarium wilt disease incidence by investigating both the external and the internal disease symptoms, and the total Fusarium wilt disease incidence was reduced to 11.7% (164 of the 1,400 plants showed disease).

Banana plants with Fusarium wilt disease were cut down, and the sites were treated with *S. morookaensis* strain Sm4-1986 again in the following year. After growing for 8 months, Fusarium wilt disease incidence of the treated banana plants was 9.1% (15 of the 164 diseased plants still remained diseased), and most of the *P. indica*-treated banana plants grown in the Sm4-1986 treated sites did not show Fusarium wilt symptoms (Supplementary Fig. 7a). In contrast, all the untreated banana plants grown in the hotspots showed severe Fusarium wilt symptoms (Supplementary Fig. 7b),

**Improved rhizosphere microbiome and soil property during the biocontrol of FWB.** We next investigated how the application of biocontrol strains change the richness and diversity of rhizosphere microbiome during the biocontrol of FWB. ACE, Chao1, and Shannon indices revealed that the rhizosphere soil of healthy plants generally harbored richer and more diverse microbial communities than that of the diseased plants, and continuous biocontrol application further increased the fungal and bacterial richness and diversity (Supplementary Table 2). To further compare the structure of microbiota in the rhizosphere between the healthy and diseased plants, we applied principal coordinate analysis (PCoA) with bray-curtis distances to analyze the microbial data. The results revealed the clusters of the microbial communities in the rhizosphere between the healthy and diseased plants were clearly separated from each other (Supplementary Fig. 8a, b), and biocontrol of FWB further improved the separation of microbial communities (Fig. 4a, b, Supplementary Fig. 8).

The linear discriminant analysis effect size (LEfSe) method was used to identify differential biomarkers in the healthy and diseased rhizosphere soils. In bacterial community, the uncultured bacteria belonging to *Acidobacteriaceae*, *Acetobacteraceae*, and *Gammaproteobacteria* were biomarkers for diseased plants, whereas *Sphingomonadaceae* and *Gemmatimonadeceae* could be used as biomarkers for healthy plants (Supplementary Fig. 9a). In fungal community, *Hydnodontaceae*, *Trechispora*, and *Morosphaeriaceae* were biomarkers for diseased plants, while *Leptodiscella*, *Acrospermales*, and *Cladorrhinum* could be used as biomarkers for healthy plants (Supplementary Fig. 9b). To gain insight into the role of important microbial species behind pathogen suppression, we identified the potential driver taxa in the microbiome networks between the healthy and diseased based on the NetShift analysis. *Chthoniobacter*, *Mesorhizobium*, *Dyella*, *Streptomyces*, and some uncultured bacteria in bacterial community (Fig. 4c) together with *Enterocarpus*, *Leptobacillium*,

*Musidium*, and *Humicola* in fungal community (Fig. 4d) were identified as the keystone taxa behind pathogen suppression in the initial microbiome of diseased plants.

To explore the relationship between soil properties and occurrence of FWB in banana plants, we measured total nitrogen (TN), total phosphorus (TP), total ferrum ($Fe^{3+}$, TF), and pH values of the soils from the healthy and diseased plants (Supplementary Table 3). Redundancy analysis (RDA) showed that higher TN, TP, and pH were positively correlated with the healthy plants, whereas higher TF was positively correlated with the diseased plants. Soil moisture had no effects on the disease (Fig. 4e, f).

**Iron is a key factor in the control of FWB.** Since high TF is always positively associated with FWB, we examined the role of iron in the control of FWB. We used ethylenediaminedi-*O*-hydroxyphenylacetic acid (EDDHA) compound, one of the most efficient iron-chelating agents, to reduce available iron in the medium[30,31]. Addition of EDDHA (final concentration of 4 mM) completely suppressed the growth of *Foc* TR4 in comparison with the control on PDA (Fig. 5a, b). On the other hand, banana plantlets treated with the same concentration of EDDHA grew well and did not show much difference to the control (Fig. 5c, d). As expected, banana plantlets infected with *Foc* TR4 showed severe disease symptoms and died (Fig. 5e), and the presence of EDDHA protected banana plantlets from being infected by *Foc* TR4 (Fig. 5f). 8-Hydroxyquinoline (8HQ) is another well-known iron chelator, and 200 μM 8HQ sufficiently suppressed *Foc* TR4 growth (Supplementary Fig. 10a, b). On the other hand, banana plantlets grown in the pots filled with *Foc* TR4-treated soil showed Fusarium disease symptoms (Supplementary Fig. 10c), but plants in pots treated with 200 μM 8HQ grew well and did not show Fusarium disease symptoms, indicating that the Fusarium wilt disease was successfully controlled (Supplementary Fig. 10d).

## Discussion

*P. indica* is a root-colonizing endophytic fungus with a broad range of host plants. It was reported that *P. indica* grew intracellularly in the root cortex but did not reach the central part of the roots when colonized barley[12,32]. However, it was observed that, in this study, *P. indica* is able to penetrate endodermis and reaches the stele of banana roots. *Foc* TR4 penetrates the cortex parenchyma of the roots and enters the xylem catheters when it infects the banana plants[29]. If they encounter in banana roots, *P. indica* is able to restrict the growth of *Foc* TR4 and reduce the disease symptoms. This hypothesis was supported by the observations that *P. indica* is able to clasp *Foc* TR4 and application of *P. indica* on banana plantlets leads to the reduced Fusarium wilt disease. Control of *Foc* TR4 growth and extension in the endophytic compartments of bananas is an important step in controlling FWB. It was reported that *P. indica* promotes plant growth by producing auxin[33,34]. The auxin-induced changes of endodermis cells are required for the initiation of lateral root primordia in underlying pericycle cells, and later the hormone auxin triggers lateral root development[35,36]. When colonized in banana roots, *P. indica* preferred to aggregate at the lateral root primordia and promote more lateral roots than untreated ones. More lateral roots enable better growing of plants, which enhances disease resistance to pathogens.

*S. morookaensis* was used to control FWB in the field. Analysis of the soil properties showed that iron increase is associated with higher incidence of FWB whereas pH increase is associated with the lower incidence of FWB, indicating that iron and pH are two important factors in the control of FWB. Consistent with these

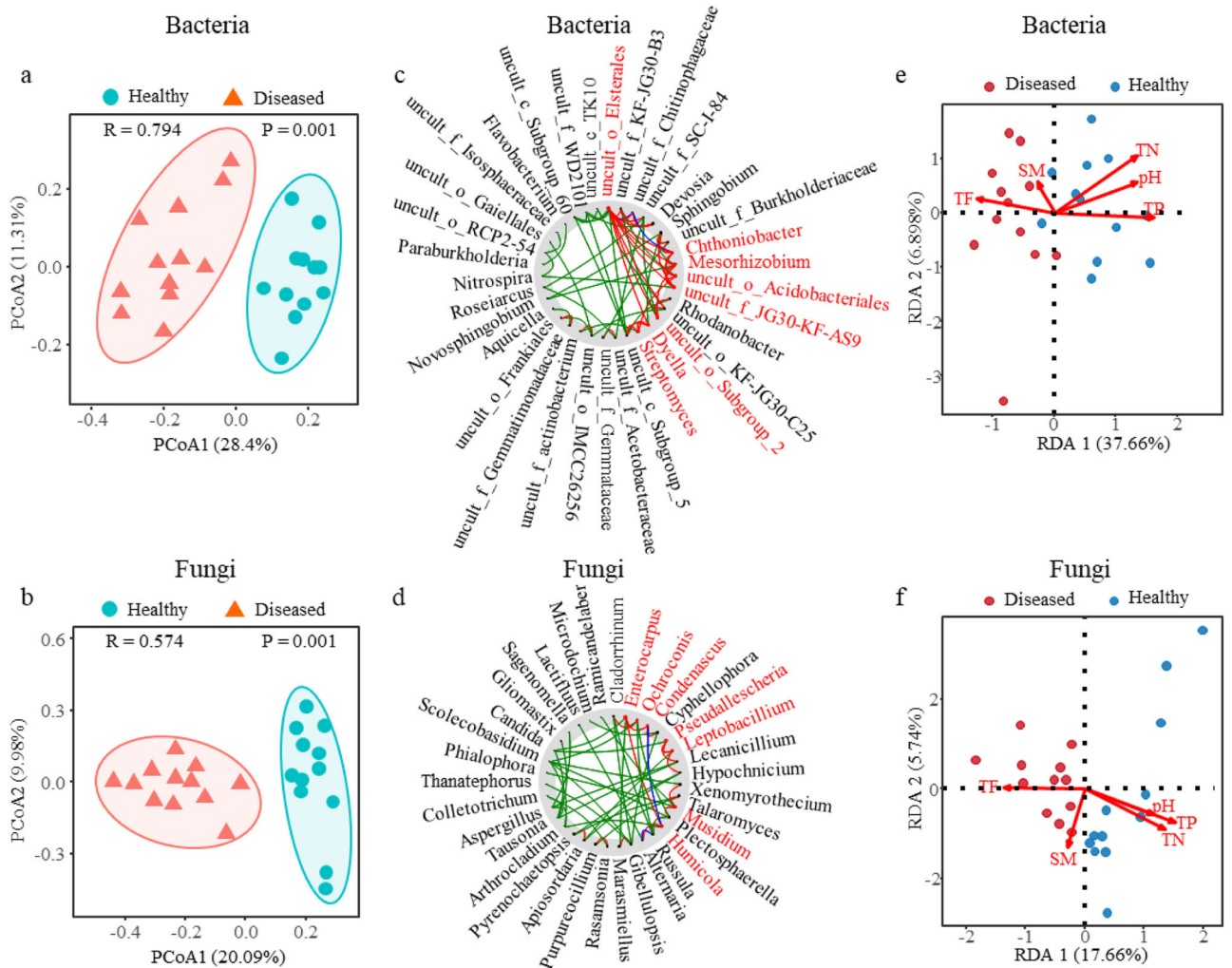

**Fig. 4 Analyses of rhizosphere microbiome and soil property. a, b** Principal coordinate analysis (PCoA) of bacterial community (**a**) and fungal community (**b**) in the rhizosphere soil between healthy ($n = 12$) and diseased ($n = 12$) plants based on the Bray-Curtis distance, showing the soil microbiomes associated with the healthy and diseased plants were clearly separated. Each symbol represents an individual. **c, d** NetShift analysis to identify potential driver taxa behind pathogen suppression based on bacterial (**c**) and fungal (**d**) networks of the rhizosphere microbiome. The node sizes are proportional to their scaled NESH (neighbor shift) scores, and a node is colored red if its betweenness increased from control to case. Large and red nodes denote particularly important driver taxa behind pathogen suppression, and the taxa names are shown in red. Edge (line) is assigned between the nodes; green edges, association present only in the diseased plant microbiome; red edges, association present only in the healthy plant microbiome; and blue, association present in both diseased and healthy plant microbiomes. **e, f** Redundancy analysis (RDA) investigating the relationship between bacterial (**e**) or fungal (**f**) communities and the soil properties. TN, total nitrogen; TP, total phosphorus; TF, total ferrum; SM, soil moisture; dots represent individual plants ($n = 12$).

results, previous reports have shown that iron competition in fungus-plant interactions is the most important mechanism for biocontrol of plant diseases caused by *Fusarium oxysporum* pathogens[37,38]. However, banana is cultivated in tropical and subtropical areas where the soils are acidified and enriched with iron, which makes it difficult to control FWB. In these respects, any strategy that increases soil pH and/or decreases iron content may help reduce the incidence of FWB. Experiments using EDDHA and 8HQ confirmed the importance of iron in the control of FWB, which establishes a causal mechanistic link between iron utilization and FWB control. Siderophores are small molecules that can easily bind to ferric iron, restricting the accessibility to other microbes, therefore, microbial strains that produce siderophores and suppress *Foc* TR4 growth are particularly attractive in the control of FWB. *S. morookaensis* strain Sm4-1986 produces different compounds that not only chelate iron but also suppress *Foc* TR4 growth. Sm4-1986 produces

various compounds, of which XcA and 6-PP play important roles in control of FWB; the former not only chelates iron but also deforms *Foc* TR4 spores, and the latter promotes plant growth and inhibits *Foc* TR4 germination. Additionally, combinatorial utilization of these two compounds synergistically increases inhibition effects on the growth of *Foc* TR4, implying the efficient inhibition of *Foc* TR4 and potential utilization of these two compounds in the control of FWB. In agreement with this, application of *S. morookaensis* strain Sm4-1986 in the field greatly reduced the incidence of FWB.

The rhizosphere microbiome greatly affects the outputs of the interaction between plants and microbes. Application of a biocontrol agent may have important impacts on the composition, structure, and functionality of the rhizosphere microbiome. The biomarker microbes were significantly different after the treatment of *S. morookaensis* strain Sm4-1986. *Acidobacteriaceae* is acidophilic and extremely abundant in acidic environments, and

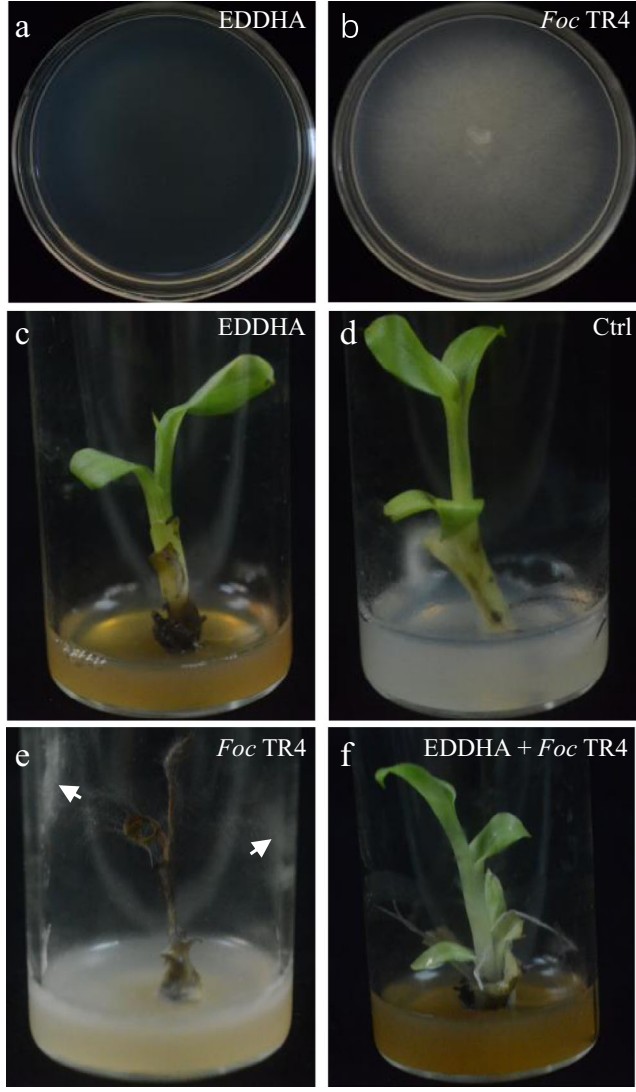

**Fig. 5 Iron-chelating suppresses *Foc* TR4 growth and inhibits Fusarium wilt disease of banana. a** Addition of EDDHA (final concentration of 4 mM) suppressed growth of *Foc* TR4. **b** *Foc* TR4 grew normally on PDA medium. **c**, **d** banana plantlets grew in ½ MS medium with (**c**) or without (**d**) 4 mM EDDHA. **e** A banana plantlet died when grew in ½ MS medium containing a small *Foc* TR4 medium plug, and the white arrows indicate *Foc* TR4 hyphae. **f** A banana plantlet grew in ½ MS medium containing 4 mM EDDHA and a small *Foc* TR4 medium plug. One representative of the plantlets ($n = 5$) in each treatment was taken a picture.

*Acetobacteraceae* can oxidize ethanol to acetic acid in neutral or acidic environments. They are two biomarkers for the diseased plants and correspond to the low pH. However, *Cladorrhinum foecundissimum* is a biomarker for the Sm4-1986-treated healthy plants, and this endophyte increases uptake of phosphorus by plants and promotes growth of the colonized plants[39]. This result indicated that *S. morookaensis* increased rhizosphere microbiome that are beneficial to banana growth and it is an efficient biocontrol agent to FWB.

Biocontrol is a comprehensive strategy that consists of many complex and interconnected factors which can influence the efficacy of biocontrol in the field. Combinatorial application of different strains is a good strategy to increase the biocontrol efficacy, but much attention must be paid to the procedures of how to apply the different strains. Various methods that deliver biocontrol strains to plants and soils also influence the

consistency of biocontrol. Thus, optimizing the mode of delivery of biocontrol strains determines the success of biocontrol. The first layer of biocontrol is to reduce the infection rate of pathogen in the soil. *S. morookaensis* produces a set of secondary compounds with different functions in suppressing *Foc* TR4 growth and reducing *Foc* TR4 spore number, which, consequently, reduces the infection chance of banana plants in the field. The secondary layer is to restrict pathogen growth and extension in the endophytic compartments of banana plants in case the pathogen escapes the first layer and colonizes the roots. *P. indica* functions in this front and inhibits *Foc* TR4 growth in banana plants. Spatiotemporal application of *P. indica* and *S. morookaensis* to banana plants and the field is to take advantages of the characteristics of these two strains to increase biocontrol efficacy of FWB.

## Methods

**Culturing microbe**. *P. indica* was propagated on PDA (potato dextrose agar). A mycelial plug (5 mm in diameter) was placed in the center of a PDA plate and cultured in an incubator at 28 °C in the dark for a week. *P. indica* on PDA plate was refreshed once a month. For liquid culture, a small mycelial plug (5 mm in diameter) was punched from mycelium margin of the stock plate and cultured in a 250 ml Erlenmeyer flask containing 100 mL PDB (potato dextrose broth) at 28 °C with a rotation speed of 200 rpm for 7 days. The culture was harvested and homogenized in a blender, and then filtered with cheese cloth. Spore number was determined using a Malassez hemocytometer and the filtrate solution was adjusted to $1 \times 10^6$ chlamydospores/ml for use. To observe lateral root formation, we grew banana plantlets in glass bottles containing 1/2 Hoagland medium[40] for a month. *P. indica* inoculum of $1 \times 10^6$ chlamydospores/ml was used to inoculate the banana plants. To observe *P. indica* colonization patterns in banana roots, we used *P. indica* solution of $1 \times 10^5$ chlamydospores/ml. For field trials, five mycelium plugs were cultured in a 1000 ml Erlenmeyer flask containing 500 ml PDB for 10 days, and then the whole culture was collected and diluted by 5 times with water for use. Banana plantlets were placed into the solution by dipping the roots for 5 sec, and then transplanted 7 days later.

*S. morookaensis* strain Sm4-1986 was propagated on PDA medium at 28 °C in the dark for a week. A mycelium plug (5 mm in diameter) from the margin of a growing colony of *S. morookaensis* strain Sm4-1986 was inoculated into 200 ml PDB in a 500 ml Erlenmeyer flask and cultured at 28 °C in a rotary shaker at 200 rpm for 10 days. The fermentation liquid was collected for banana plantlet inoculation. For field trials, five mycelium plugs (1 cm in diameter) were inoculated into 500 ml PDB in a 1000 ml Erlenmeyer flask and cultured at 28 °C in a rotary shaker at 200 rpm for 10 days. The whole fermentation broth was collected and diluted by 3 times with water and then applied to the field.

*Foc* TR4 and a GFP-tagged *Foc* TR4 displayed similar growth characteristics and virulence to bananas were used[41]. *Foc* TR4 was used for inoculating banana plantlets and GFP-tagged *Foc* TR4 for observing the fluorescence. *Foc* TR4 or GFP-tagged *Foc* TR4 was cultured on PDA at 28 °C in the dark for 7 days. A mycelium plug (5 mm in diameter) from the margin of a growing colony of *Foc* TR4 or GFP-tagged *Foc* TR4 was put in a 250 ml Erlenmeyer flask containing 100 mL PDB and cultured at 28 °C with a rotation speed of 200 rpm for 5 days. The culture was filtered with cheese cloth, and the number of conidia in the filtrate was counted using a Malassez hemocytometer and adjusted to $1 \times 10^6$ conidia/ml for use.

**Banana material**. Cavendish cultivar 'Brazilian' banana plants (*Musa acuminate* L. AAA group)[42], which were purchased from the Center of Tissue Culture, South China Botanical Garden, Guangzhou, China. Banana plantlets at different growth stages were used in different experiments.

**Compound isolation and structure elucidation**. *S. morookaensis* strain 4-1986 was fermented in PDB on a large-scale at 28 °C for a week with a rotation speed of 200 rpm. Xerucitrinin A, 6-pentyl-α-pyrone, and other compounds were isolated from the fermentation broth. *Streptomyces morookaensis* strain Sm4-1986 were fermented in PDB medium and cultured at 200 rpm 28 °C for a week. The supernatants were collected, dried, and exhaustively extracted with EtOAc for four times. The extracted layers were evaporated to yield a dark brown gum, which was eluted with a gradient mixture of *n*-hexane/EtOAc (from 20:1 to 1:1, V/V) by silica gel column chromatography. TLC analysis separated twenty fractions, and different fractions were then subjected to different column chromatography elution. The purified fractions were determined by UV, IR, NMR, and mass spectroscopic analyses, and compound classification and structure elucidation were determined[26].

**Interaction between *P. indica* and *Foc* TR4**. *P. indica* and *Foc* TR4 were co-cultured on a glass slide, which was first covered by a thin layer of PDA. A mycelium plug of *P. indica* was placed on one end of the glass slide that was

covered by a petri dish which was then cultured in an incubator at 28 °C for 3 days. After that, a Foc TR4 plug was placed on the other end of the glass slide, and the petri dish was covered and incubated at 28 °C for another 4 days. During the period of cultivation, the leading hyphae of P. indica and Foc TR4 could meet somewhere between the two plugs. A small disc containing the overlapped hyphae of P. indica and Foc TR4 was sliced out from the place where they met and then subjected to scanning electron microscope observation[43]. Hyphae from the individually cultured P. indica and Foc TR4 were used as controls.

**Effects of XcA and 6-PP on morphology of Foc TR4 spores**. A plug of Foc TR4 mycelia was cultured in 50 ml PDB in a 100 ml Erlenmeyer flask shaken in a rotary shaker with 200 rpm at 28 °C for 5 days. The culture was filtered using sterile gauze, and the filtrate was centrifuged at 1000 rpm for 5 min to collect conidia, which were then resuspended in PDB and adjusted to $1 \times 10^7$ conidia/ml. Aliquot 200 μl of the conidial suspension into the cells of a 96-well plate. XcA or 6-PP was added to the cells containing Foc TR4 conidia to make a final concentration of 3 mM for XcA and 0.96 mM for 6-PP. Each treatment was repeated three times. After 72 h incubation at 28 °C in darkness; conidia were subjected to scanning electron microscope observation[44].

**Effects of 6-PP on ultrastructure of Foc TR4 spores**. Foc TR4 mycelia were cultured in 1000 ml PDB at 28 °C for 5 days. The culture was filtrated with sterile gauze, and the filtrate centrifuged at 1000 rpm for 5 min to collect conidia, which were resuspended in 1000 ml PDB and adjusted to $1 \times 10^7$ conidia/ml. 6-PP was added to the culture making a final concentration of 6-PP at 0.96 mM. The conidial culture was incubated at 28 °C for 24 h in darkness; centrifuged at 3000 rpm for 10 min at 4 °C to collect spores[45]. The collected conidia were subjected to transmission electron microscope observation.

**Effects of XcA and 6-PP on growth of banana plantlets**. Micro-propagated banana plantlets were grown on 1/2 MS in glass tubes containing either 3 mM XcA or different concentrations of 6-PP. The glass tubes were put in the tissue culture room at 26 °C with a photoperiod of 14/10 h light/dark. Banana plantlets were observed and photographed at the indicated time.

**Scanning electron microscopy**. Conidial samples or mycelial discs were fixed in a solution containing 2.5% glutaraldehyde and 2% paraformaldehyde for 2 h, and then subjected to vacuum pumping to let samples sink to the bottom of tube. After that, the samples were stored in a refrigerator for 12 h. Pouring off the solution and washed samples three times with 0.1 M phosphate buffer (80 g NaCl, 32.3 g Na₂HPO4·12H₂O, and 4.5 g NaH₂PO4·2H₂O in 1000 ml ddH₂O, pH 7.2) at 4 °C; 40 min per washing. Washed samples were vapor-fixed with 1% (w/v) aqueous osmium tetroxide for 2.5 h at room temperature, followed by washing three times with 0.1 M phosphate buffer at 4 °C, and each time for 5 min. After washing, samples were subjected to gradient dehydration in different concentrations of ethanol as in the order of 30%, 50%, 70%, 80%, and 90% at 4 °C; each for 10 min, and finally kept in 100% ethanol for 50 min. At the end of dehydration, samples were dried in a critical point dryer (Leica EM CPD300, Germany). The dried samples were sputter-coated with gold palladium in a sputter coaster (Leica EM ACE600, Germany)[45]. Conidial or hyphae morphology of Foc TR4 or P. indica was observed by SEM (JSM-6360LV, Japan).

**Transmission electron microscopy**. 6-PP-treated Foc TR4 spores were fixed in the solution containing 2.5% glutaraldehyde and 2% paraformaldehyde for 3 h. The samples were washed four times for 15 min with 0.1 M phosphate buffer (80 g NaCl, 32.3 g, Na₂HPO4·12H₂O, and 4.5 g NaH₂PO4·2H₂O in 1000 ml ddH₂O, pH 7.2); and washed twice with 0.1 M phosphate buffer; 30 min for each. Washed samples were post-fixed in 1.0% osmium tetraoxide (in 0.1 M phosphate buffer) for 4 h. The fixed samples were washed six times again with 0.1 M phosphate buffer for a period of 2 h. During this period, the washing buffer was changed every 15 min for the first four washing times and 30 min for the last two times. After washing, samples were pre-stained with 0.5% uranyl acetate (in 0.1 M phosphate buffer) for overnight. All the above steps were carried out at 4 °C.

On the next day, spore samples were washed six times with 0.1 M phosphate buffer at 4 °C; 15 min for the first 4 times and 30 min for the last two times. After washing, samples were dehydrated in gradient ethanol solutions (30%, 50%, 70%, 80%, 90%) at room temperature (RT) for 20 min each, and final in 100% ethanol for 30 min with one change. After dehydration, samples were passed in three changes of epoxypropane: epikote 812 (3:1 for 30 min, 1:1 for 60 min, and 1:3 for 150 min) at RT, and then in epikote 812 at RT for 3 h. Finally, samples were kept in fresh epikote 812 for overnight. In next day, samples were kept in fresh epikote 812 at RT for 7 h, and then were picked up to put in a small plastic box containing epikote 812. Samples were hardened in an oven at 60 °C for 12 h. Finally, the hardened blocks containing samples were sectioned into ultrathin sections of about 70 nm[46], which were observed under a transmission electron microscope (Tecnai G2 SpiriBio TWIN).

**Colonization pattern of P. indica in banana roots**. Banana plantlets were cultured in 1/2 Hoagland medium[40] containing P. indica at the concentration of $1 \times 10^5$ chlamdospores/ml for two weeks. Banana roots were harvested, washed thoroughly with running tap water, and cut into 1 cm segments, which were stained with 0.01% acid fuchsin-lactic acid for 5 min and destained in lactic acid for 1 min. These were sectioned with an automatic vibrating slicing machine (Leica VT1200S, Germany);[47] and sections were observed under an optical microscope (Leica DMI3000, Germany) and images were taken.

**Greenhouse experiments**. Experiments with banana plantlets in pots were carried out in greenhouse at 25 °C/18 °C (day/night) with a photoperiod of 14 h /10 h (light/dark). Prior to inoculation, all plantlets were maintained in greenhouse for a week to adapt environments. Plantlets with similar size were selected for inoculation. Each treatment contained 15 plantlets and was repeated three times. To inoculate plantlets, 50 ml inoculum were used for each pot. P. indica of $1 \times 10^6$ chlamydospores/ml or Foc TR4 of $1 \times 10^6$ conidia/ml was used for inoculation. 8QH was used at the concentration of 200 μM. After inoculation, banana plantlets were observed at the indicated time.

**Field trials**. The field trials were carried out in a land of 0.8 hectares in Longmen County (23°72'77"N, 114°25'49"E), Huizhou City, Guangdong Province, China. The small farmer had planted banana (Brazilian variety) in the land for more than 10 years and recently abandoned it because of the high incidence of Fusarium wilt caused by Foc TR4. Fifteen plants in a line were treated as a block, and the treatment and control blocks were arranged alternately in field. Banana transplantation holes of the treatment block were treated with S. morookaensis strain 4-1986 a week before transplantation. On the other hand, banana plants at the seven-leaf stage were treated with P. indica by submerging the roots into the inoculation solution. Banana transplantation was carried out in late February 2019. In the following days, standard irrigation and fertilization practices were applied to banana plants, and other managements were followed the normal farm operations in banana orchard. At the end of the year, external symptoms of Fusarium wilt disease were investigated[48]. Plants showing external symptom of leaf chlorosis were cut down, and the pseudostems were chopped off to examine the internal symptoms[49]. The sites were treated with S. morookaensis strain 4-1986 again. New banana plants were treated with P. indica and transplanted into the treated hotspots in 2020. Fusarium wilt disease incidence of bananas plants was investigated at the end of the year. External and internal symptoms were examined to determine whether the plants were actually infected by Foc TR4.

**Soil property analysis**. Soil samples were collected from the evenly distributed four sites around a plant. pH values of each sample (soil material: distilled water = 1: 10, w/v) were measured three times using a pH meter (PHS-25, Shanghai Inesa Instrument Co. Ltd., China). Total nitrogen (TN), total phosphorus (TP), and total ferrum (TF) were detected using an Elemental analyzer (PE2400, PerkinElmer, USA).

**Rhizosphere microbial Illumina MiSeq sequencing**. Soils were collected from four locations evenly distributed around the plants. The bacterial community composition of the rhizosphere was assessed by sequencing the V3-V4 region of the 16 S rRNA gene using the universal primers 338 F/806 R (forward primers 5′-ACTCCTACGGGAGGCAGCA-3′ and reverse primers 5′-GGACTACGCGG-TATCTAAT-3′). The fungal ITS1 region was amplified using primer set ITS1 (5′-GGAAGTAAAAGTCGTAACAAGG-3′) and ITS2 (5′-GCTGCGTTCTTCATC-GATGC-3′)[50]. The PCR reactions were carried out in a 50 μL reaction mixture containing 1.5 μL each primer, 1 μL dNTP, 10 μL Buffer, 0.2 μL Q5 High Fidelity DNA Polymerase, 10 μL High GC Enhancer, and 40 ng soil DNA template. The PCR conditions for bacteria were initiated at 95 °C for 3 min, followed by 25 cycles of denaturation at 95 °C for 45 s, annealing at 50 °C for 45 s, and extension at 68 °C for 90 s, followed by a final elongation at 68 °C for 7 min, and then held at 4 °C. The PCR conditions for fungi were initiated at 98 °C for 2 min, followed by 30 cycles of denaturation at 98 °C for 30 s, annealing at 50 °C for 30 s, and extension at 72 °C for 1 min, followed by a final elongation at 72 °C for 5 min, and then held at 4 °C. The PCR products were pooled and visualized on 1.8% agarose gels, purified using a MinElute® PCR Purification Kit according to the manufacturer's instructions, and quantified using QuantiFluorTM-ST (Promega, USA).

High-throughput sequencing was carried out on the Illumina MiSeq platform (BioMarker Technologies Co. Ltd, China). After pyrosequencing, raw sequences were processed with Prinseq (PRINSEQlite 0.19.5) to remove low-quality data and improve the syncretic rates of the subsequent sequence. Split sequences for each sample were merged using FLASH V1.2.7[51]. Using Usearch with a cut-off of 97% similarity, the OTUs were clustered and the taxonomic classification was performed using RDP Classifier (Version 2.2, based on Bergey's taxonomy) with the classification threshold set at 0.8. The sequences were taxonomically identified by a BLASTn search of a curated NCBI database.

**Statistics and reproducibility**. Experiments with banana plantlets in pots were carried out in greenhouse at 25 °C/18 °C (day/night) with a photoperiod of 14 h /10 h (light/dark). Prior to inoculation, all plantlets were maintained in greenhouse

for a week to adapt to environments. Plantlets with similar size were selected for inoculation. Each treatment contained 15 plantlets and was repeated three times. The field trials were carried out in a land of 0.8 hectares in Longmen County (23°72′77″N, 114°25′49″E), Huizhou City, Guangdong Province, China. Fifteen plants in a line were treated as a block, and the treatment and control blocks were arranged alternately in field.

All data analyses were analyzed using the SPSS 20.0 program (SPSS Inc., USA), and the significance between control and treatments was assigned at $P < 0.05$ using a one-way analysis of variance (ANOVA) with LSD test. Alpha diversity indices, including Chao 1, ACE, and Shannon, were calculated using the OTU table in QIIME[51].

The soil microbiome composition was ordinated by principal coordinates analysis (PCoA) using bray-curtis distance. Differences between microbiome composition of healthy and diseased plants were calculated by using PERMANOVA and ANOSIM. Bray–Curtis distances are sensitive to rare OTUs and thus emphasize differences in the presence or absence of taxa[52]. R package (v3.2.0) was used to draw the graph of redundancy analysis (RDA).

Linear discriminant analysis effect size (LEfSe) was used to explore the most differentiated OTUs between health and diseased conditions[53]. Two screening criteria were used: (1) linear discriminant analysis with a score of ≥ 3.0 (healthy condition relative to diseased condition); (2) significance test with $P < 0.05$.

We also used the NetShift method to identify potential keystone driver taxa based on differences in network interactions between healthy and diseased plant microbiome (https://web.rniapps.net/netshift)[54].

**Reporting summary**. Further information on research design is available in the Nature Portfolio Reporting Summary linked to this article.

## Data availability

All sequence data generated in this study have been deposited in NCBI SRA database and the accession numbers are reported in Supplementary Data 1. All source data underlying the graphs and charts presented in the figures uploaded as Supplementary Data 2. Any remaining information can be obtained from the corresponding author upon reasonable request.

## Code availability

All code or algorithms used in this study are published and referenced in the Methods.

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

## Acknowledgements

The authors thank Prof. Kai-Wun Yeh from Institute of Plant Biology of National Taiwan University for providing *P. indica*, and Dr. Gan-Jun Yi from Fruit Research Institute of Guangdong Academy of Agricultural Sciences for providing GFP-tagged *Foc* TR4 and *Foc* TR4. This work was supported by CAS grant Regional Key Project of Science and Technology Service Network Initiative from Chinese Academy of Sciences (No. KFJ-STS-QYZX-044), and grant of Science and Technology Commissioner for Rural Revitalization from Guangdong Province to J.X.L.

## Author contributions

J.X.L. designed the study, collected data, analyzed results, and wrote the paper. Z.Y.Z performed experiments, collected data, and analyzed results. G.Y.W isolated compounds and identified structures of xerucitrinin A and 6-pentyl-α-pyrone. Z.Y.Z and Y.F.D performed laser confocal microscope and scanning electron microscope. Z.Y.Z and X.Y.H performed transmission electron microscope. H.B.T identified compound structure and analyzed results. Y.P.C analyzed results. Z.H.T designed the study.

## Competing interests

The authors declare no competing interests.
