## [Peer Review File · Communications Biology]

Reviewers' comments:

Reviewer #1 (Remarks to the Author):

GVNakato

**Spatiotemporal biocontrol and rhizosphere microbiome analysis of**
**Fusarium wilt of banana**

Zhiyan Zhu^{1,2}, Guiyun Wu^{3,4}, Rufang Deng⁵, Xiaoying Hu⁵, Haibo Tan⁴, Yaping
Chen², Zhihong Tian^{1,*}, Jianxiong Li^{6,*}

¹College of Life Sciences, Yangtze University, Jingzhou, China.

²South China Botanical Garden, Chinese Academy of Sciences, Guangzhou, China

³Science and Technology Innovation Center, Guangzhou University of Chinese
Medicine, Guangzhou, China;

⁴Key Laboratory of Plant Resources Conservation and Sustainable Utilization, South
China Botanical Garden, Chinese Academy of Sciences, Guangzhou China;

⁵Public Laboratory of Sciences, South China Botanical Garden, Chinese Academy of
Sciences, Guangzhou, China

⁶Guangxi Key Laboratory of Agro-environment and Agric-products safety, College of
Agriculture, Guangxi University, Nanning, China

*Corresponding author: email: jxli920@gxu.edu.cn (J.X.L); zhtian@yangtzeu.edu.cn
(Z.H.T)

**Abstract**

The soil-borne fungus *Fusarium oxysporum* f. sp. *cubense* tropical race 4 (*Foc* TR4)
causes Fusarium wilt of banana (FWB), which devastates banana production
worldwide. Biocontrol is considered to be the most efficient approach to reducing
FWB. Here we introduce an approach that spatiotemporally apply *Piriformospora*
*indica* and *Streptomyces morookaensis* strain Sm4-1986 according to their respective
strength to increase biocontrol efficacy of FWB. *P. indica* successfully colonizes
banana roots and promotes lateral root formation, and it also inhibits *Foc* TR4 growth
inside the banana plants and reduces FWB. *S. morookaensis* strain Sm4-1986 secretes
different secondary compounds, of which xerucitrinin A (XcA) and 6-pentyl- α -pyrone
(6-PP) show the strongest anti-*Foc* TR4 activity. XcA is able to chelate iron, an
essential nutrient in pathogen-plant interaction that determines the output of FWB. 6-
PP, a volatile organic compound, inhibits *Foc* TR4 germination and promotes banana
growth. Biocontrol trials in the field demonstrated that application of *S. morookaensis*
improves soil properties and increases soil rhizosphere-associated microbes that are
beneficial to banana growth, and thus significantly reduces disease incidence of FWB.
Our study suggests that optimal utilization of the different biocontrol strains increases
efficacy of biocontrol and that operating the iron accessibility in the rhizosphere is a
promising strategy to control FWB.

**Introduction**

Bananas (*Musa* ssp.), originated in Southeast Asia and the Western Pacific^{1,2}, are
now widely distributed throughout the humid tropics and sub-tropics where they
provide a staple food for about 400 million people in the developing countries in Africa,
Asia, and Latin America³. Bananas are the most exported fruit in the world, having
production of 129 million tons and export trade value of 13.6 billion dollars in 2019⁴.

Fusarium wilt of banana (FWB) caused by the soil-borne fungus *Fusarium*
*oxysporum* f. sp. *cubense* (*Foc*) is one of the most destructive disease in banana
production worldwide, which has restricted banana production for more than a century⁵.
The epidemic of FWB leads to the almost complete replacement of the *Foc* race 1
susceptible Gros Michel with the resistant Cavendish, which currently covers ca. 40%
of the global production and may be well the only banana present on supermarket
shelves of non-producer countries^{3,6}. However, a newly emerged race of *Foc*, tropical
race 4 (TR4), is virulent not only on Cavendish but also on all other banana cultivars.
*Foc* TR4 caused serious losses in banana plantation worldwide, which resulted in
abandonment of thousands of hectares of banana orchards. Currently, there are no
effective methods to control FWB caused by *Foc* TR4.

The *Foc* pathogen can linger in soil for up to 30 years even in the absence of plant
hosts, which makes it particularly difficult to be eliminated from the infected soil^{7,8}. As
being a vascular pathogen, *Foc* colonizes banana roots and reaches the vascular
bundles³, leading to the ineffectiveness of chemical control. Repeated use of chemical
reagents has raised great concern for environmental pollution and human health.
Breeding resistant cultivars is thought to be the most effective way to control FWB, but
all commercial banana varieties tested are susceptible to *Foc* TR4, and they are
propagated by cloning due to the nature of sterile triploid⁹. Thus, biological control of
FWB has gained great interest¹⁰.

*Piriformospora indica* is a well-known endophytic fungus that colonizes roots of a
broad spectrum of plant species and confers diverse beneficial effects on host plants by
promoting growth or enhancing disease resistance¹¹⁻¹³. Besides, it also acts as a

[revised manuscript text omitted]

Since XcA and 6-PP were isolated from Sm4-1986 and each has antifungal activity
against *Foc* TR4, we hypothesized that coexistence of these two compounds may have
synergistic effect on suppressing *Foc* TR4. To test this hypothesis, we used response
surface methodology (RSM) to analyze the interaction between XcA and 6-PP and to
examine how these two compounds synergistically affect *Foc* TR4 germination at
different concentrations. RSM analyses revealed that coexistence of XcA and 6-PP at
lower concentrations was able to suppress *Foc* TR4 spore germination, indicating a
synergistic effect of these two compounds (Fig. 2h and Extended Data Fig. 2). Given
the inhibitory effect of XcA and 6-PP on *Foc* TR4, we therefore asked what are their
effects on banana growth. Experiments with banana plantlets showed that lower
concentrations of 6-PP (<150 μ M) promoted banana plantlet growth although higher
concentrations (>200 μ M) showed side effects to banana plantlets (Extended Data Fig.
3). Regarding to XcA, banana plantlets grew well when treated for 65 days by 3 mM
XcA, the concentration to inhibit *Foc* TR4 germination, when compared with the
control. This indicates that XcA may be non-toxic to banana plants (Extended Data Fig.
4).

***P. indica* induces lateral root formation and suppresses *Foc* TR4 growth.** *P. indica*
is symbiotically associated with a variety of host plants²⁵. To explore the colonization
pattern of *P. indica* in banana, we observed the *P. indica*-treated banana roots under
microscope. *P. indica* entered banana roots primarily through root hairs (Extended

Data Fig. 5a) and, later, crossed cortex and endodermis, and then moved to stele and
aggregated at the lateral root primordium initiation sites (Fig. 3a and Extended Data
Fig. 5b). In agreement with these phenomena, *P. indica*-treated banana plantlets
showed more lateral roots than untreated ones (Fig. 3b, c).

*Foc* TR4 is able to penetrate cortical parenchyma cells and reach the vascular
bundle tissues of roots²⁶. Therefore, inhibiting *Foc* TR4 growth and extension in the
endophytic compartments of banana roots is an important part of FWB control. To
investigate the interaction between *Foc* TR4 and *P. indica*, we co-cultured these two
strains and observed their overlaid hyphae. SEM images showed that *P. indica* tightly
clasped *Foc* TR4 and resulted in the collapse of *Foc* TR4 hyphae, suggesting an
inhibitory effect of *P. indica* on *Foc* TR4 (Fig. 3d-f).

We then examined the effects of *P. indica* on growth of banana and control of
Fusarium wilt disease. Inoculation with *P. indica* (1×10^6 chlamydospores/ml)
promoted the growth of banana plantlets (Extended Data Fig. 6a, b and Supplementary
Table 1). On the contrary, infection with *Foc* TR4 led to the occurrence of typical
Fusarium wilt disease symptoms on banana plantlets (Extended Data Fig. 6c). However,
if the banana plantlets were first inoculated with *P. indica* and then infected by *Foc* TR4,
they showed less disease symptoms and grew better than *Foc* TR4-treated plantlets
(Extended Data Fig. 6c, d and Supplementary Table 1). In agreement with the external
symptoms, investigation of the internal symptoms of banana rhizomes showed that *P.*
*indica* treatment greatly reduced the discoloration caused by *Foc* TR4 infection
(Extended Data Fig. e-h).

[revised manuscript text omitted]

sufficiently suppressed *Foc* TR4 growth (Extended Data Fig. 10a, b). On the other hand,
banana plantlets grown in the pots filled with *Foc* TR4-treated soil showed Fusarium
disease symptoms (Extended Data Fig. 10c), but those plants in the pots that were
treated with 200 μ M 8HQ grew well and did not show Fusarium disease symptoms,
indicating that the Fusarium wilt disease was successfully controlled (Extended Data
Fig. 10d).

Discussion

*P. indica*, a versatile root endophytic symbiont, grows intracellularly in the root
cortex but not in the central part of the roots beyond endodermis when associated with
barley^{11,29}, however it is able to penetrate endodermis and reach the stele of banana
roots. *Foc* TR4 penetrates the cortex parenchyma of the roots and enters the xylem
catheters when it infects the banana plants²⁶. If they encounter in banana roots, *P. indica*
is able to restrict the growth of *Foc* TR4 and reduce the disease symptoms. This
hypothesis was supported by the observations that *P. indica* is able to clasp *Foc* TR4
and application of *P. indica* on banana plantlets leads to the reduced Fusarium wilt
disease. Control of *Foc* TR4 growth and extension in the endophytic compartments of
bananas is an important step of the procedure for controlling FWB, and *P. indica*

[revised manuscript text omitted]

- 14. Harrach, B.D., Baltruschat, H., Barna, B., Fodor, J. & Kogel, K. H. The
Mutualistic fungus *Piriformospora indica* protects barley roots from a loss of
antioxidant capacity caused by the necrotrophic pathogen *Fusarium culmorum*.
*Mol. Plant Microbe Interact.* **26**, 599-605 (2013).
- 15. Sun, C. et al. The beneficial fungus *Piriformospora indica* protects Arabidopsis
from *Verticillium dahliae* infection by downregulation plant defense responses.

- *BMC Plant Biol.* **14**, 268-282 (2014).
- 16. Rose, S., Parker, M. & Punja, Z. K. Efficacy of biological and chemical treatments
for control of Fusarium root and stem rot on greenhouse cucumber. *Plant Dis.* **87**,
1462-1470 (2003).
- 17. Olanrewaju, O. S. & Babalola, O. O. *Streptomyces*: implications and interactions
in plant growth promotion. *Appl. Microbiol. Biotechnol.* **103**, 1179-1188 (2019).
- 18. Jing, T. et al. Newly isolated *Streptomyces* sp. JBS5-6 as a potential biocontrol
agent to control banana Fusarium wilt: genome sequencing and secondary
metabolite cluster profiles. *Front. Microbiol.* **11**, e602591 (2020).
- 19. Zhang, H. Y., Xue, Q. H., Shen, G. H. & Wang, D. S. Effects of actinomycetes
agent on ginseng growth and rhizosphere soil microflora. *Chin. J. Appl. Ecol.* **8**,
2287-2293 (2013).
- 20. Hyakumachi, M. Studies on biological control of soilborne plant pathogens. *J. Gen.*
*Plant Pathol.* **66**, 272-274 (2000).
- 21. Gu, S. et al. Competition for iron drives phytopathogen control by natural
rhizosphere microbiomes. *Nat. Microbiol.* **5**, 1002-1010 (2020).
- 22. Zhu, Z., Tian, Z. & Li, J. A *Streptomyces morookaensis* strain promotes plant
growth and suppresses Fusarium wilt of banana. *Trop. Plant Pathol.* **46**, 175-185
(2020).
- 23. Wu, G. Y. et al. Chemical constituents from the *Streptomyces morookaensis* strain
Sm4-1986. *Nat. Prod. Res.* <https://doi:10.1080/14786419.2021.1881095> (2021).
- 24. Liu, S. et al. Fusaric acid instigates the invasion of banana by *Fusarium oxysporum*
f. sp. *ubense* TR4. *New Phytol.* **225**, 913-929 (2020).
- 25. Unnikumar, K. R., Sree, K. S. & Varma, A. *Piriformospora indica*: a versatile root
endophytic symbiont. *Symbiosis* **60**, 107-113 (2013).
- 26. Dong, H. et al. Histological and gene expression analyses in banana reveals the
pathogenic differences between races 1 and 4 of banana Fusarium wilt pathogen.
*Phytopathology* **109**, 1029-1042 (2019).
- 27. Hernandez-Apaolaza, L. & Lucena, L. L. Influence of irradiation time and solution

- concentration on the photochemical degradation of EDDHA/Fe³⁺: effect of its
photodecomposition products on soybean growth. *J. Sci. Food Agric.* **91**, 2024-
2030 (2011).
- 28. Kovacs, K. et al. Revisiting the iron pools in cucumber roots: identification and
localization. *Planta* **244**, 167-169 (2016).
- 29. Deshmukh, S. et al. The root endophytic fungus *Piriformospora indica* requires
host cell death for proliferation during mutualistic symbiosis with barley. *Proc.*
*Natl Acad. Sci. USA* **103**, 18450–18457 (2006).
- 30. Sirrenberg, A. et al. *Piriformospora indica* affects plant growth by auxin
production. *Physiol. Plantarum* **131**, 581-589 (2007).
- 31. Varma, A., Kost, G. & Oelmüller, R. *Piriformospora indica* (Springer-Verlag
Berlin Heidelberg, 2013).
- 32. Vermeer, J. E. M. et al. A spatial accommodation by neighboring cells is required
for organ initiation in Arabidopsis. *Science* **343**, 178-183 (2014).
- 33. Wachsman, G., Zhang, J. Y., Moreno-Risueno, M. A., Anderson, C. T. & Benfey, P.
422 N. Cell wall remodeling and vesicle trafficking mediate the root clock in
Arabidopsis. Preprint at *bioRxiv* <https://doi.org/10.1101/2020.03.10.985747> (2020).
- 34. Scher, F. M. & Baker, R. Effect of *Pseudomonas putida* and a synthetic iron
chelator on induction of soil suppressiveness of Fusarium wilt pathogens.
*Phytopathology* **72**, 1567-1573 (1982).
- 35. Lopez-Berges, M. S. et al. Iron competition in fungus-plant interactions: The battle
takes place in the rhizosphere. *Plant Signal. Behav.* **8**, e23012 (2013).
- 36. Gasoni, L. & de Gurfinkel, B.S. The endophyte *Cladorrhinum foecundissimum* in
cotton roots: phosphorus uptake and host growth. *Mycol. Res.* **101**, 867-870 (1997).

**Methods**

**Culturing microbe.** *Piriformospora indica* was a gift from Prof. Kai-Wun Yeh of
Institute of Plant Biology of National Taiwan University, Taiwan. *Streptomyces*
*morookaensis* strain Sm4-1986 was purchased from China General Microbiological

Culture Collection Center (CGMCC# 4.1986), Beijing, China. GFP-tagged *Foc* TR4
was a gift from Dr. Gan-Jun Yi of Fruit Research Institute of Guangdong Academy of
Agricultural Sciences, Guangdong, China.

[revised manuscript text omitted]

Linear discriminant analysis was used to explore the most discriminating OTUs
between health and diseased conditions using LEfSe⁵⁰. Two screening criteria were
used: (1) linear discriminant analysis with a score of ≥ 3.0 (healthy condition relative
to diseased condition) and (2) significance test with $P < 0.05$.

We also used the NetShift method to identify potential keystone driver taxa based
on differences in network interactions between healthy and diseased plant microbiome
(<https://web.rniapps.net/netshift>)⁵¹.

**Accession numbers for sequence data.** Accession numbers of the rhizosphere microbiome

data determined at the end of the field experiment in 2019 (33 samples, paired end reads).

treatment	Bacterial			Fungi		
	Accession no	Paired end read 1	Paired end read 2	Accession no	Paired end read 1	Paired end read 2
Diseased	SAMN27606370	abDiseased1_1.fq	abDiseased1_2.fq	SAMN27606453	afDiseased1_1.fq	afDiseased1_2.fq
Diseased	SAMN27606371	abDiseased2_1.fq	abDiseased2_2.fq	SAMN27606454	afDiseased2_1.fq	afDiseased2_2.fq
Diseased	SAMN27606372	abDiseased3_1.fq	abDiseased3_2.fq	SAMN27606455	afDiseased3_1.fq	afDiseased3_2.fq
Healthy	SAMN27606373	abHealthy1_1.fq	abHealthy1_2.fq	SAMN27606456	afHealthy1_1.fq	afHealthy1_2.fq
Healthy	SAMN27606374	abHealthy2_1.fq	abHealthy2_2.fq	SAMN27606457	afHealthy2_1.fq	afHealthy2_2.fq
Healthy	SAMN27606375	abHealthy3_1.fq	abHealthy3_2.fq	SAMN27606458	afHealthy3_1.fq	afHealthy3_2.fq
Biotreated	SAMN27606376	abBiotreated1_1.fq	abBiotreated1_2.fq	SAMN27606459	afBiotreated1_1.fq	afBiotreated1_2.fq
Biotreated	SAMN27606377	abBiotreated2_1.fq	abBiotreated2_2.fq	SAMN27606460	afBiotreated2_1.fq	afBiotreated2_2.fq
Biotreated	SAMN27606378	abBiotreated3_1.fq	abBiotreated3_2.fq	SAMN27606461	afBiotreated3_1.fq	afBiotreated3_2.fq
Diseased	SAMN27605964	abA01_1.fq	abA01_2.fq	SAMN27606065	afA01_1.fq	afA01_2.fq
Diseased	SAMN27605965	abA02_1.fq	abA02_2.fq	SAMN27606066	afA02_1.fq	afA02_2.fq
Diseased	SAMN27605966	abA03_1.fq	abA03_2.fq	SAMN27606067	afA03_1.fq	afA03_2.fq
Diseased	SAMN27605967	abA04_1.fq	abA04_2.fq	SAMN27606068	afA04_1.fq	afA04_2.fq
Diseased	SAMN27605968	abA05_1.fq	abA05_2.fq	SAMN27606069	afA05_1.fq	afA05_2.fq
Diseased	SAMN27605969	abA06_1.fq	abA06_2.fq	SAMN27606070	afA06_1.fq	afA06_2.fq
Diseased	SAMN27605970	abA07_1.fq	abA07_2.fq	SAMN27606071	afA07_1.fq	afA07_2.fq
Diseased	SAMN27605971	abA08_1.fq	abA08_2.fq	SAMN27606072	afA08_1.fq	afA08_2.fq
Diseased	SAMN27605972	abA09_1.fq	abA09_2.fq	SAMN27606073	afA09_1.fq	afA09_2.fq
Diseased	SAMN27605973	abA10_1.fq	abA10_2.fq	SAMN27606074	afA10_1.fq	afA10_2.fq
Diseased	SAMN27605974	abA11_1.fq	abA11_2.fq	SAMN27606075	afA11_1.fq	afA11_2.fq
Diseased	SAMN27605975	abA12_1.fq	abA12_2.fq	SAMN27606076	afA12_1.fq	afA12_2.fq
Healthy	SAMN27605976	abB01_1.fq	abB01_2.fq	SAMN27606077	afB01_1.fq	afB01_2.fq
Healthy	SAMN27605977	abB02_1.fq	abB02_2.fq	SAMN27606078	afB02_1.fq	afB02_2.fq
Healthy	SAMN27605978	abB03_1.fq	abB03_2.fq	SAMN27606079	afB03_1.fq	afB03_2.fq
Healthy	SAMN27605979	abB04_1.fq	abB04_2.fq	SAMN27606080	afB04_1.fq	afB04_2.fq
Healthy	SAMN27605980	abB05_1.fq	abB05_2.fq	SAMN27606081	afB05_1.fq	afB05_2.fq
Healthy	SAMN27605981	abB06_1.fq	abB06_2.fq	SAMN27606082	afB06_1.fq	afB06_2.fq
Healthy	SAMN27605982	abB07_1.fq	abB07_2.fq	SAMN27606083	afB07_1.fq	afB07_2.fq
Healthy	SAMN27605983	abB08_1.fq	abB08_2.fq	SAMN27606084	afB08_1.fq	afB08_2.fq
Healthy	SAMN27605984	abB09_1.fq	abB09_2.fq	SAMN27606085	afB09_1.fq	afB09_2.fq
Healthy	SAMN27605985	abB10_1.fq	abB10_2.fq	SAMN27606086	afB10_1.fq	afB10_2.fq
Healthy	SAMN27605986	abB11_1.fq	abB11_2.fq	SAMN27606087	afB11_1.fq	afB11_2.fq
Healthy	SAMN27605987	abB12_1.fq	abB12_2.fq	SAMN27606088	afB12_1.fq	afB12_2.fq

Accession numbers of the rhizosphere microbiome data determined at the end of the field
 experiment in 2020 (33 samples, paired end reads).

treatment	Bacterial			Fungi		
	Accession no	Paired end read 1	Paired end read 2	Accession no	Paired end read 1	Paired end read 2
Diseased	SAMN27606490	bbDiseased1_1.fq	bbDiseased1_2.fq	SAMN27606507	bfDiseased1_1.fq	bfDiseased1_2.fq
Diseased	SAMN27606491	bbDiseased2_1.fq	bbDiseased2_2.fq	SAMN27606508	bfDiseased2_1.fq	bfDiseased2_2.fq
Diseased	SAMN27606492	bbDiseased3_1.fq	bbDiseased3_2.fq	SAMN27606509	bfDiseased3_1.fq	bfDiseased3_2.fq
Healthy	SAMN27606493	bbHealthy1_1.fq	bbHealthy1_2.fq	SAMN27606510	bfHealthy1_1.fq	bfHealthy1_2.fq
Healthy	SAMN27606494	bbHealthy2_1.fq	bbHealthy2_2.fq	SAMN27606511	bfHealthy2_1.fq	bfHealthy2_2.fq
Healthy	SAMN27606495	bbHealthy3_1.fq	bbHealthy3_2.fq	SAMN27606512	bfHealthy3_1.fq	bfHealthy3_2.fq
Biotreated	SAMN27606496	bbBiotreated1_1.fq	bbBiotreated1_2.fq	SAMN27606513	bfBiotreated1_1.fq	bfBiotreated1_2.fq
Biotreated	SAMN27606497	bbBiotreated2_1.fq	bbBiotreated2_2.fq	SAMN27606514	bfBiotreated2_1.fq	bfBiotreated2_2.fq
Biotreated	SAMN27606498	bbBiotreated3_1.fq	bbBiotreated3_2.fq	SAMN27606515	bfBiotreated3_1.fq	bfBiotreated3_2.fq
Diseased	SAMN27606096	bbA01_1.fq	bbA01_2.fq	SAMN27606338	bfA01_1.fq	bfA01_2.fq
Diseased	SAMN27606097	bbA02_1.fq	bbA02_2.fq	SAMN27606339	bfA02_1.fq	bfA02_2.fq
Diseased	SAMN27606098	bbA03_1.fq	bbA03_2.fq	SAMN27606340	bfA03_1.fq	bfA03_2.fq
Diseased	SAMN27606099	bbA04_1.fq	bbA04_2.fq	SAMN27606341	bfA04_1.fq	bfA04_2.fq
Diseased	SAMN27606100	bbA05_1.fq	bbA05_2.fq	SAMN27606342	bfA05_1.fq	bfA05_2.fq
Diseased	SAMN27606101	bbA06_1.fq	bbA06_2.fq	SAMN27606343	bfA06_1.fq	bfA06_2.fq
Diseased	SAMN27606102	bbA07_1.fq	bbA07_2.fq	SAMN27606344	bfA07_1.fq	bfA07_2.fq
Diseased	SAMN27606103	bbA08_1.fq	bbA08_2.fq	SAMN27606345	bfA08_1.fq	bfA08_2.fq
Diseased	SAMN27606104	bbA09_1.fq	bbA09_2.fq	SAMN27606346	bfA09_1.fq	bfA09_2.fq
Diseased	SAMN27606105	bbA10_1.fq	bbA10_2.fq	SAMN27606347	bfA10_1.fq	bfA10_2.fq
Diseased	SAMN27606106	bbA11_1.fq	bbA11_2.fq	SAMN27606348	bfA11_1.fq	bfA11_2.fq
Diseased	SAMN27606107	bbA12_1.fq	bbA12_2.fq	SAMN27606349	bfA12_1.fq	bfA12_2.fq
Healthy	SAMN27606108	bbB01_1.fq	bbB01_2.fq	SAMN27606350	bfB01_1.fq	bfB01_2.fq
Healthy	SAMN27606109	bbB02_1.fq	bbB02_2.fq	SAMN27606351	bfB02_1.fq	bfB02_2.fq
Healthy	SAMN27606110	bbB03_1.fq	bbB03_2.fq	SAMN27606352	bfB03_1.fq	bfB03_2.fq
Healthy	SAMN27606111	bbB04_1.fq	bbB04_2.fq	SAMN27606353	bfB04_1.fq	bfB04_2.fq
Healthy	SAMN27606112	bbB05_1.fq	bbB05_2.fq	SAMN27606354	bfB05_1.fq	bfB05_2.fq
Healthy	SAMN27606113	bbB06_1.fq	bbB06_2.fq	SAMN27606355	bfB06_1.fq	bfB06_2.fq
Healthy	SAMN27606114	bbB07_1.fq	bbB07_2.fq	SAMN27606356	bfB07_1.fq	bfB07_2.fq
Healthy	SAMN27606115	bbB08_1.fq	bbB08_2.fq	SAMN27606357	bfB08_1.fq	bfB08_2.fq
Healthy	SAMN27606116	bbB09_1.fq	bbB09_2.fq	SAMN27606358	bfB09_1.fq	bfB09_2.fq
Healthy	SAMN27606117	bbB10_1.fq	bbB10_2.fq	SAMN27606359	bfB10_1.fq	bfB10_2.fq
Healthy	SAMN27606118	bbB11_1.fq	bbB11_2.fq	SAMN27606360	bfB11_1.fq	bfB11_2.fq
Healthy	SAMN27606119	bbB12_1.fq	bbB12_2.fq	SAMN27606361	bfB12_1.fq	bfB12_2.fq

**References**

[revised manuscript text omitted]

i hyphae of *Foc* TR4. Scale bars, 2 μm.

n

d

i

c

a

s

t

i

m

u

l

a

t

e

s

l

a

t

e

**Fig. 4 Analyses of rhizosphere microbiome and soil property.** a and b, Principal

coordinate analysis (PCoA) of bacterial community (a) and fungal community (b) in

the rhizosphere soil between healthy ($n = 12$) and diseased ($n = 12$) plants based on

the Bray-Curtis distance, showing the soil microbiomes associated with the healthy

and diseased plants were clearly separated. Each symbol represents an individual. c

and d, NetShift analysis to identify potential driver taxa behind pathogen suppression

based on bacterial (c) and fungal (d) networks of the rhizosphere microbiome. The

node sizes are proportional to their scaled NESH (neighbor shift) scores, and a node is

colored red if its betweenness increased from control to case. Large and red nodes

denote particularly important driver taxa behind pathogen suppression, and the taxa

names are shown in red. Edge (line) is assigned between the nodes; green edges,

association present only in the diseased plant microbiome; red edges, association

present only in the healthy plant microbiome; and blue, association present in both

diseased and healthy plant microbiomes. e and f, Redundancy analysis (RDA)
investigating the relationship between bacterial (e) or fungal (f) communities and the
soil properties. TN, total nitrogen; TP, total phosphorus; TF, total ferrum; SM, soil
moisture; dots represent individual plants ($n = 12$).

**Fig. 5 Iron-chelating suppresses *Foc* TR4 growth and inhibits Fusarium wilt**

**disease of banana.** a, Addition of EDDHA (final concentration of 4 mM) suppressed

growth of *Foc* TR4. b, *Foc* TR4 grew normally on PDA medium. c and d, banana

plantlets grew in ½ MS medium with (c) or without (d) 4 mM EDDHA. e, A banana

plantlet died when grew in ½ MS medium containing a small *Foc* TR4 medium plug,

and the white arrows indicate *Foc* TR4 hyphae. f, A banana plantlet grew in ½ MS

medium containing 4 mM EDDHA and a small *Foc* TR4 medium plug. One
representative of the plantlets ($n = 5$) in each treatment was taken a picture.

Reviewer #2 (Remarks to the Author):

In this work authors studied the biocontrol mechanism of microbiome analysis against Fusarium wilt of banana (FWB) using advanced methods such GFP-tagged, TEM, SEM, and Illumina MiSeq sequencing. What I think is an interesting job, as a contribution to the solution of FW, a global issue in banana production systems. But I think in results section, it would make it easier to understand by extrapolate the results to the field settings and thus would make more sense. It would also be important to explain how the results impact small scale farmer's matrix in terms of production/ yield/ economy etc. where authors conducted field trails. Also, as authors mentioned that they got Piriformospore indica as a gift from Taiwan and Streptomyces morookaensis was purchased from Microbiological Culture Collection Center, Beijing, China which I feel require more work to examine the adaptability of these foreign strains in local soils. How long they will survive, what is the shelf life and impact on indigenous beneficial microbial communities. Similarly soil physico-chemical and biological properties should be studied more in detail and not only pH, total nitrogen (TN), total phosphorus (TP), and total ferrum (TF) and effects with microbial culture inoculations.

Manuscript review

Title: Spatiotemporal biocontrol and rhizosphere microbiome analysis of Fusarium wilt of banana

The manuscript is an interesting account of efforts towards exploring the use of *P. indica* and *S. morookaensis* as potential biocontrol agents in managing Fusarium wilt Tropical Race 4 through promotion of vigorous plant growth and secretion of secondary compounds respectively.

The manuscript title is reflective of the main ideas within the article. The introduction is systematically written and the significance of the manuscript is well explained. Although the results and conclusions are accurate and supported by the content, the authors lack a systemic presentation of this section. They need to write it out more elaborately and succinctly. The methods used in this study are appropriate. References need to be synchronized. Overall, this work is original and logical.

Specific comments were inserted directly in the manuscript.

REVIEWERS' COMMENTS:

Reviewer #2 (Remarks to the Author):

Many thanks for revisions, accepted in current form.